# Association between kidney measurements and cognitive performance in patients with ischemic stroke

**Chunyan Zhang**[1]*, **Guofang Xue**[1], **Yanjuan Hou**[2], **Pengfei Meng**[1], **Huizhong Gao**[1], **Bo Bai**[1], **Dongfang Li**[1]

1 Department of Neurology, Second Hospital, Shanxi Medical University, Taiyuan, Shanxi, China,
2 Department of Nephrology, Second Hospital, Shanxi Medical University, Taiyuan, Shanxi, China

* marzipanzcy@126.com

## Abstract

### Background

Individuals with chronic kidney disease (CKD) are at a substantially higher risk for stroke, which may predispose individuals to cognitive impairment. However, the association of low estimated glomerular filtration rate (eGFR) and albuminuria with poorer cognitive performance in patients with stroke is not fully understood, and the current evidence for this association is contradictory. Our aim was to retrospectively investigate whether low eGFR and albuminuria, as indicated by the urine albumin-creatinine ratio (UACR), are independently or jointly associated with worse cognitive performance in patients with ischemic stroke.

### Methods

This retrospective study included 608 patients with acute ischemic stroke. Their UACR and eGFR values were obtained from inpatient medical records. Global cognitive function was assessed with the mini-mental state exam (MMSE) and Montreal Cognitive Assessment (MoCA) one month after hospital discharge. The relationship between renal measures and cognitive performance was assessed using univariate and multiple linear regression analyses. Potential confounders included age, gender, BMI, education, diabetes and hypertension history, NIHSS score, smoking and alcohol consumption status, serum total cholesterol, triglyceride, fasting glucose, uric acid, homocysteine, systolic blood pressure, and either eGFR or UACR.

### Results

Patients had an average age of 66.6±4.1 years, and 48% were females. Average eGFR and UACR were 88.4±12.9 ml/min/1.73m$^2$ and 83.6±314.2 mg/g, respectively. The number of patients with eGFR $\geq$90, 60–89, and <60 ml/min/1.73 m$^2$ was 371 (61%), 207 (34%), and 30 (5%), respectively, and the percentage of patients with UACR <30 mg/g, 30–300 mg/g, and >300 mg/g was 56%, 39%, and 5%, respectively. Multivariate adjusted models showed that eGFR was independently associated with MMSE (β = -0.4; 95% CI = -0.5,-0.4; p

**Data Availability Statement:** All relevant data are within the paper and its Supporting Information files.

**Funding:** This study was supported by the Scientific Research Foundation for Doctors, the Second Hospital of Shanxi Medical University, China (Grant No: 201601-9), Natural Science Foundation of Shanxi Province (Grant No: 201801D221411 and 20210302124427), Science and Technology Department of Shanxi Province, China. None of the authors have received a salary from the funders. Funders play a role in the collection, analysis, and preparation of the research data.

**Competing interests:** The authors have declared that no competing interests exist.

<0.001) and MoCA (β = -0.6; 95% CI = -0.7,-0.5; p <0.001). However, UACR was not significantly correlated with MMSE or MoCA.

## Conclusion

In patients with ischemic stroke, reduced eGFR but not albuminuria was associated with lower cognitive performance. These results show that the eGFR decline could be an effective indicator of cognitive impairment after a stroke. Therefore, regular monitoring and early detection of mild renal dysfunction in patients with acute ischemic stroke might be needed.

## Introduction

Globally, cognitive impairment is a major health problem during old age. Due to better living conditions and healthcare, humans have extended lifespans, and this has led to a rapid increase in the prevalence of cognitive dysfunction disorders [1].

Chronic kidney disease (CKD) is a global disease burden and a major health problem that has a high economic cost [2]. CKD is defined by indicators of renal dysfunction, such as estimated glomerular filtration rate (eGFR), and kidney damage, such as urine albumin-creatinine ratio (UACR) [3]. Numerous cross-sectional or longitudinal studies have been performed to explore the relationship between kidney disease and cognitive dysfunction, but the available data are conflicting [4–15].

A recent study on the Northern Ireland Cohort for the Longitudinal Study of Ageing (NICOLA) revealed that decreased eGFR had a significant association with an increased risk of cognitive dysfunction in a cohort of older adults [16]. The Maastricht Study found albuminuria but not eGFR to be independently associated with cognitive impairment, specifically within the domain of information processing speed. However, both albuminuria and eGFR were more significantly associated with cognitive impairment in older individuals [6]. A prospective study in a Chinese population aged 80 years and above found early-stage CKD to have a correlation with cognitive decline [17]. Palmer et al. found that in the African-American population with type 2 diabetes (T2D), eGFR and albuminuria were associated with cognitive dysfunction, even in mild kidney disease [18]. Cognitive impairment was found to occur early in CKD [19]. However, Helmer et al. did not find an increased risk of cognitive decline or dementia to be associated with low eGFR level. Rather, they found that a faster decline in eGFR and proteinuria was associated with incidence of cognitive impairment and dementia with a vascular component [20].

Post-stroke cognitive impairment (PSCI) is common after stroke and occurs in up to 60% of stroke survivors in the first year after stroke, with a higher rate seen shortly after stroke [21]. The incidence of cognitive impairment after cerebrovascular accidents (CVA) in China is as high as 80.9%, including 48.91% for cognitive impairment without dementia and 32.05% for cognitive impairment with dementia [22]. Patients with CVA are at a high risk of suffering cognitive dysfunction. CVA increased the relative risk of dementia by 3.7-fold to 6.6-fold in older adults aged 61–74 years, and advanced the onset of dementia by nearly 10 years [23]. The clinical determinants of PSCI are not fully understood. Various putative risk factors have been inconsistently reported, including older age, risk factors for cerebrovascular disease (e.g., hypertension, diabetes mellitus, and atrial fibrillation), stroke features, and lesion characteristics [24]. The kidney and the brain have strong similarities in vascular organization, and a complex interplay has been found between CKD and cerebrovascular disease [25]. Individuals

with CKD are at a substantially higher risk for stroke, which may predispose individuals to cognitive impairment [26]. However, only a few studies have examined the association of low eGFR and albuminuria with poorer cognitive performance in individuals at high risk or patients with stroke, and the current evidence for such an association is contradictory. A longitudinal study found that CKD was independently related to incident dementia in patients with vascular risk factors and partly supported the role of common vascular mechanisms in the underlying pathophysiology of AD dementia [27]. Recently, a study in patients with TIA and stroke found that CKD was not independently associated with either pre- or post-event dementia, and this suggests that renal-specific mechanisms are unlikely to play an important role in the etiology of CKD [28].

The identification of modifiable risk factors of cognitive decline after stroke is of vital importance in early preventive and intervention strategies for PSCI [21, 26, 29]. Furthermore, kidney deficits may have a potential association with stroke and post-stroke cognitive impairment. Therefore, this study aimed to investigate whether low eGFR and albuminuria, as measured by UACR, are independently or jointly associated with worse cognitive performance in patients with ischemic stroke.

## Materials and methods

### Data source

Using a retrospective design, this study analyzed data from 608 patients aged 60 to 80 years with acute ischemic stroke within 72 h of onset between January 2016 and December 2020. Their data were obtained from inpatient medical records and a database of cognitive assessment results for outpatient visits one month after hospital discharge. Data were collected from September 1, 2019. During or after data collection, investigators had access to information that could identify individual patients. None of the patients had a history of cognitive disorder before stroke and were able to complete the assessment. Patients who were unable to undergo the examination, such as those with aphasia or consciousness disorder, were excluded. The study was conducted at the Department of Neurology, Second Hospital of Shanxi Medical University, Taiyuan, a tertiary hospital in Shanxi, China. This study was approved by the Ethics Committee of the Second Hospital of Shanxi Medical University (approval no. 2019YX214). The research was conducted in accordance with the Helsinki Declaration. Due to the retrospective nature of the study, and the data being anonymized, informed consent from the patients was not required. The data included renal function parameters, such as serum creatinine and UACR, and cognitive function test findings, such as mini-mental status examination (MMSE) and Montreal Cognitive Assessment (MoCA) scores.

### Clinical and laboratory data

Clinical data extracted from hospital records on admission included systolic and diastolic blood pressure on admission, self-reported demographic characteristics (age, gender, height, weight, years of education, smoking and alcohol consumption status, and diabetes mellitus and hypertension history), laboratory results (total cholesterol, triglyceride, homocysteine, uric acid, fasting glucose, urea, creatinine, and UACR). Additionally, data on neural function assessment (National Institutes of Health Stroke Scale [NIHSS] score) were available. All patients were diagnosed with large artery atherosclerosis (LAA), cardioembolism (CE), small artery occlusion (SAO), stroke of other determined cause (SOC), or stroke of undetermined cause (SUC), according to the TOAST classification [30]. Body mass index (BMI) was calculated as weight (in kilograms) divided by height (in meters) squared. eGFR was calculated using the Chronic Kidney Disease-Epidemiology Collaboration (CKD-EPI) equation, which is

more accurate than the modification of diet in renal disease (MDRD) equation [31].

$$\text{eGFR} = 141 \times \min(\text{Scr}/k, 1)^{\alpha} \times \max(\text{Scr}/k, 1)^{-1.209} \times 0.993^{\text{age}} \times 1.018[\text{if female}]$$

Where serum creatinine (Scr) is expressed in mg/dL and age in years; $\kappa$ is 0.7 for females and 0.9 for males; $\alpha$ is -0.329 for females and -0.411 for males; min indicates the minimum of Scr/$\kappa$ or 1, and max indicates the maximum of Scr/$\kappa$ or 1.

## Cognitive impairment testing

All the cognitive function tests were assessed by two neuropsychological assessors trained in consistency using MMSE and MoCA one month after hospital discharge. The MMSE and MoCA scores ranged from 0 to 30; higher scores indicated better cognitive function. The highest possible MMSE score was 30 points, which was scored across four dimensions: 10 for orientation, 6 for registration and memory, 5 for attention and calculation, and 9 for language and visuospatial abilities. The MoCA domains included visuospatial and executive function (5 points), naming (3 points), memory (5 points), attention (6 points), language/abstraction (5 points), and orientation (6 points).

## Statistical analysis

The study was based on analysis of retrospectively obtained data. Patient characteristics were presented as mean ± standard deviations (SD) for continuous variables, and categorical variables were reported as frequencies and percentages. Differences between groups for continuous data were tested by one-way analysis of variance (ANOVA). Differences between groups for proportions were tested with a Chi-square test. Data on UACR were not normally distributed; therefore, they were log-transformed before analysis. Standard definitions were used to categorize eGFR data into three groups, comprising ≥90 ml/min/1.73 m$^2$ as reference, 60–89 ml/min/1.73 m$^2$, and <60 ml/min/1.73 m$^2$. Thereafter, eGFR was analyzed as a continuous variable (per 5 mL/min/1.73 m$^2$ decreased). UACR values were categorized as <30 mg albumin/g creatinine (normal to mildly increased as reference), 30–300 mg/g (moderately increased), and >300 mg/g (severely increased). Multivariate linear regression was used to assess the association between eGFR or UACR and cognitive performance, with eGFR or UACR as the independent variable and MoCA and MMSE scores as the dependent variables. Two models, Models 1 and 2, were used for the regression analysis. Model 1 was crude, and multivariate-adjusted Model 2 was constructed to adjust for potential confounding variables associated with kidney function and/or cognitive function, including age, gender, BMI, education, diabetes and hypertension history, NIHSS score, stroke subtypes, smoking and alcohol consumption status, serum total cholesterol, triglyceride, uric acid, homocysteine, fasting glucose, systolic blood pressure (SBP), and either eGFR or UACR. Crude and adjusted ß with 95% CIs are presented. Two-tailed $P$ <0.05 was considered statistically significant. All analyses were performed using Empower (R) (www.empowerstats.com, X&Y Solutions Inc., Boston, MA) and R (http://www.R-project.org).

## Results

### Population characteristics

Table 1 shows demographic and laboratory data on all 608 patients. Patients had a mean age of 66.6 years (SD: 4.1 years) and 48% of patients were females. Average eGFR and UACR were 88.4±12.9 ml/min/1.73 m$^2$ and 83.6 ± 314.2 mg/g, respectively. Levels of eGFR were >90 ml/min/1.73m$^2$ (N = 371, 61%), 60–89 ml/min/1.73m$^2$ (N = 207, 34%), and <60 ml/min/1.73m$^2$

**Table 1. Demographic and clinical characteristics of patients stratified by eGFR and UACR.**

| | Overall | Estimated glomerular filtration(ml/min/1.73 m$^2$) | | | P value | Urinary ACR (mg/g) | | | P value |
|---|---|---|---|---|---|---|---|---|---|
| | | ≥90 | 60–89 | <60 | | <30 | 30–300 | >300 | |
| | n = 608 | n = 371 | n = 207 | n = 30 | | n = 338 | n = 240 | n = 30 | |
| Age, y | 66.6 ± 4.1 | 65.5 ± 3.5 | 68.0 ± 4.5 | 69.8 ± 4.5 | <0.001 | 66.4 ± 4.2 | 66.8 ± 4.1 | 66.3 ± 3.7 | 0.45 |
| Male, n(%) | 316 (52.0) | 192 (51.8) | 105 (50.7) | 19 (63.3) | 0.43 | 186 (55.0) | 115 (47.9) | 15 (50.0) | 0.235 |
| BMI,[a] kg/m$^2$ | 25.0 ± 3.6 | 24.8 ± 3.4 | 24.9 ± 3.8 | 26.6 ± 4.0 | 0.033 | 24.6 ± 3.4 | 25.3 ± 3.7 | 25.9 ± 3.8 | 0.019 |
| Education (> 6 years), n (%) | 200 (32.9) | 130 (35.0) | 58 (28.0) | 12 (40.0) | 0.158 | 113 (33.4) | 77 (32.1) | 10 (33.3) | 0.943 |
| Systolic blood pressure on admission, mmHg | 171.9 ± 20.2 | 170.0 ± 19.8 | 174.2 ± 20.7 | 178.9 ± 19.4 | 0.008 | 170.3 ± 18.4 | 173.8 ± 22.1 | 174.8 ± 23.3 | 0.079 |
| Diastolic blood pressure on admission, mmHg | 92.9 ± 12.2 | 92.1 ± 12.1 | 93.9 ± 12.4 | 96.5 ± 12.7 | 0.057 | 91.5 ± 11.7 | 94.6 ± 12.7 | 95.5 ± 12.8 | 0.006 |
| Smoking, n (%) | | | | | 0.112 | | | | 0.057 |
| Never | 355 (58.5) | 229 (61.7) | 114 (55.3) | 12 (40.0) | | 182 (54.0) | 153 (63.7) | 20 (66.7) | |
| Former | 64 (10.5) | 33 (8.9) | 26 (12.6) | 5 (16.7) | | 34 (10.1) | 26 (10.8) | 4 (13.3) | |
| Current | 188 (31.0) | 109 (29.4) | 66 (32.0) | 13 (43.3) | | 121 (35.9) | 61 (25.4) | 6 (20.0) | |
| Alcohol drinking, n (%) | | | | | 0.474 | | | | 0.18 |
| Never | 389 (64.0) | 245 (66.0) | 128 (61.8) | 16 (53.3) | | 205 (60.7) | 163 (67.9) | 21 (70.0) | |
| Former | 52 (8.6) | 27 (7.3) | 21 (10.1) | 4 (13.3) | | 33 (9.8) | 15 (6.2) | 4 (13.3) | |
| Current | 167 (27.5) | 99 (26.7) | 58 (28.0) | 10 (33.3) | | 100 (29.6) | 62 (25.8) | 5 (16.7) | |
| Diabetes mellitus,[b] n (%) | 50 (8.2) | 34 (9.2) | 13 (6.3) | 3 (10.0) | 0.45 | 21 (6.2) | 23 (9.6) | 6 (20.0) | 0.019 |
| Hypertension, n (%) | 361 (59.4) | 215 (58.0) | 127 (61.4) | 19 (63.3) | 0.656 | 189 (55.9) | 154 (64.2) | 18 (60.0) | 0.138 |
| Stroke subtypes | | | | | 0.149 | | | | 0.911 |
| Large artery | 281 (46.2) | 183 (49.3) | 80 (38.6) | 18 (60.0) | | 156 (46.2) | 110 (45.8) | 15 (50.0) | |
| Small vessel occlusion | 309 (50.8) | 176 (47.4) | 121 (58.5) | 12 (40.0) | | 171 (50.6) | 124 (51.7) | 14 (46.7) | |
| Cardioembolism | 13 (2.1) | 9 (2.4) | 4 (1.9) | 0 (0.0) | | 8 (2.4) | 4 (1.7) | 1 (3.3) | |
| Other determined etiology | 3 (0.5) | 1 (0.3) | 2 (1.0) | 0 (0.0) | | 1 (0.3) | 2 (0.8) | 0 (0.0) | |
| Undetermined etiology | 2 (0.3) | 2 (0.5) | 0 (0.0) | 0 (0.0) | | 2 (0.6) | 0 (0.0) | 0 (0.0) | |
| Laboratory results | | | | | | | | | |
| Total cholesterol, mmol/L | 5.7 ± 1.2 | 5.6 ± 1.1 | 5.8 ± 1.3 | 5.8 ± 1.3 | 0.055 | 5.7 ± 1.2 | 5.7 ± 1.2 | 6.0 ± 1.1 | 0.423 |
| Triglyceride, mmol/L | 1.6 ± 0.9 | 1.7 ± 1.1 | 1.5 ± 0.7 | 1.6 ± 0.7 | 0.036 | 1.5 ± 0.8 | 1.8 ± 1.1 | 1.6 ± 0.5 | 0.016 |
| Homocysteine, umol/L | 16.1 ± 9.4 | 14.7 ± 7.9 | 17.0 ± 8.6 | 27.9 ± 18.7 | <0.001 | 15.5 ± 7.1 | 17.0 ± 11.7 | 17.1 ± 10.2 | 0.124 |
| Uric Acid, mmol/L | 298.6 ± 75.4 | 284.5 ± 68.9 | 312.3 ± 75.2 | 377.7 ± 89.6 | <0.001 | 299.0 ± 71.8 | 296.2 ± 79.5 | 313.0 ± 81.9 | 0.511 |
| Fasting glucose, mmol/L | 6.1 ± 1.8 | 6.2 ± 2.0 | 6.1 ± 1.6 | 6.3 ± 1.4 | 0.698 | 6.0 ± 1.7 | 6.2 ± 1.9 | 6.7 ± 2.5 | 0.082 |
| Estimated glomerular filtration rate, mL/min/1.73m$^2$ | 88.4 ± 12.9 | 96.0 ± 5.3 | 79.8 ± 8.5 | 53.2 ± 8.2 | <0.001 | 89.0 ± 11.9 | 88.3 ± 13.0 | 81.8 ± 19.8 | 0.013 |
| Log10 Urinary ACR | 1.5 ± 0.5 | 1.5 ± 0.5 | 1.5 ± 0.4 | 1.7 ± 0.7 | 0.246 | 1.2 ± 0.2 | 1.8 ± 0.3 | 2.8 ± 0.3 | <0.001 |
| Neural function assessment, mean (SD) | | | | | | | | | |
| MMSE score, mean ± SD (range 0–30) | 21.4 ± 2.1 | 22.0 ± 1.9 | 21.1 ± 1.6 | 17.5 ± 2.8 | <0.001 | 21.5 ± 2.0 | 21.4 ± 2.1 | 20.8 ± 2.9 | 0.18 |
| MoCA score, mean ± SD (range 0–30) | 17.9 ± 2.3 | 18.7 ± 2.0 | 17.2 ± 1.9 | 13.4 ± 2.6 | <0.001 | 18.0 ± 2.3 | 17.9 ± 2.3 | 17.2 ± 2.8 | 0.18 |
| NIHSS score, mean ± SD (range 0–24) | 5.0 ± 2.2 | 5.1 ± 2.3 | 4.8 ± 2.1 | 5.6 ± 1.8 | 0.091 | 5.1 ± 2.2 | 4.8 ± 2.1 | 5.3 ± 2.8 | 0.302 |

Abbreviations: BMI, body mass index. UACR: urine albumin-creatinine ratio

Values are presented as mean ± SD for continuous variables and n (%) for categorical variables.

[a]BMI was calculated as weight in kilograms divided by height in meters squared.

[b]Diabetes mellitus was defined as self-reported physician diagnosed diabetes.

[C]Hypertension was defined as self-reported physician diagnosed hypertension.

(N = 30, 5%). The number of patients with UACR <30 mg/g was 338 (56%). Moderately increased UACR, defined as UACR 30–300 mg/g, was found in 240 (39%) patients, and severely increased UACR, defined as UACR >300 mg/g, was detected in 30 (5%) patients.

Patients with lower eGFR categories were more likely to be older and former or current alcohol consumers and smokers. In addition, they were more likely to have diabetes mellitus, and hypertensive disease, and high systolic blood pressure (SBP) and diastolic blood pressure (DBP) on admission. Their homocysteine, blood glucose, uric acid, and total cholesterol levels were more likely to be higher, as well as a higher BMI. The eGFR levels were lower in patients with albuminuria than in normal patients. As compared with patients with UACR <30 mg/g, patients with UACR ≥30 mg/g were more likely to be living with obesity, diabetes mellitus, and hypertensive disease. Additionally, they were more likely to have higher SBP and DBP, blood glucose, uric acid, and homocysteine levels. Patients had lower cognitive scores in the lower eGFR and higher UACR categories (Table 1).

## Independent and combined associations between eGFR and UACR levels and cognitive performance

Table 2 contains the results of unadjusted and fully adjusted (age, gender, BMI, education, diabetes and hypertension history, NIHSS score, stroke subtypes, smoking and alcohol consumption status, serum total cholesterol, triglyceride, uric acid, homocysteine, fasting glucose, SBP, and eGFR or UACR) models for continuous and categorical relationships between measures of renal disease and cognitive tests. The correlation between UACR and MMSE and MoCA scores in crude and multivariate adjustment were statistically non-significant. In the crude and adjusted analyses, no independent associations were observed between UACR, as a category, and cognitive performance. Similarly, no independent associations were observed when UACR was analyzed as a continuous variable in Models 1 and 2 (Table 2).

The association between eGFR and the tests of cognitive performance was evaluated with crude and multivariable linear regression analyses. eGFR was analyzed as a categorical (≥90, 60–90, and <60 ml/min/1.73 m$^2$) and a continuous (per 5 ml/min/1.73 m$^2$ decrease) variable. In the multivariable adjustment model, for patients with eGFR 60–90 ml/min/1.73 m$^2$ compared to patients with eGFR ≥90 ml/min/1.73 m$^2$, a positive association existed between eGFR and MMSE scores (ß,-0.7; 95%CI, -1.0 to -0.3) and MoCA scores (ß,-1.4; 95%CI, -1.7 to -1.0).

**Table 2. Linear regression analysis of the association between levels of eGFR and albuminuria measured by UACR and cognitive performance.**

| Variables | No. of patients (%) | MMSE | | | | MoCA | | | |
|---|---|---|---|---|---|---|---|---|---|
| | | Crude | | adjusted* | | Crude | | adjusted* | |
| | | ß (95%CI) | P | ß (95%CI) | P | ß (95%CI) | P | ß (95%CI) | P |
| eGFR, (ml/min/1.73m$^2$) categories | | | | | | | | | |
| ≥90 | 371(61) | 0 | | 0 | | 0 | | 0 | |
| 60–90 | 207(34) | -0.9 (-1.2, -0.6) | <0.001 | -0.7 (-1.0, -0.3) | <0.001 | -1.5 (-1.8, -1.2) | <0.001 | -1.4 (-1.7, -1.0) | <0.001 |
| <60 | 30(5) | -4.4 (-5.1, -3.8) | <0.001 | -4.2 (-5.0, -3.4) | <0.001 | -5.3 (-6.0, -4.5) | <0.001 | -5.2 (-6.1, -4.4) | <0.001 |
| eGFR per 5-unit decline | 608(100) | -0.4 (-0.5, -0.4) | <0.001 | -0.4 (-0.5, -0.4) | <0.001 | -0.5 (-0.6, -0.5) | <0.001 | -0.6 (-0.7, -0.5) | <0.001 |
| UACR(mg/g) categories | | | | | | | | | |
| <30 | 338(56) | 0 | | 0 | | 0 | | 0 | |
| 30–300 | 240(39) | -0.1 (-0.4, 0.3) | 0.716 | 0.0 (-0.3, 0.3) | 0.971 | -0.0 (-0.4, 0.3) | 0.858 | 0.0 (-0.3, 0.3) | 0.826 |
| ≥300 | 30(5) | -0.7 (-1.5, 0.0) | 0.064 | -0.3 (-1.0, 0.4) | 0.387 | -0.8 (-1.7, 0.0) | 0.065 | -0.2 (-0.9, 0.6) | 0.68 |
| log-UACR | 608(100) | -0.2 (-0.6, 0.2) | 0.262 | -0.0 (-0.4, 0.3) | 0.784 | -0.3 (-0.6, 0.1) | 0.214 | -0.0 (-0.4, 0.3) | 0.834 |

Abbreviations: CI: confidence interval; eGFR: estimated glomerular filtration rate; UACR: urine albumin-creatinine ratio

*Adjusted for age, gender, BMI, education, diabetes and hypertension history, NIHSS score, stroke subtypes, smoking and alcohol drinking status, serum total cholesterol, triglyceride, uric acid, homocysteine, fasting glucose, systolic blood pressure and either baseline estimated glomerular filtration rate or UACR.

The positive association between eGFR <60 ml/min/1.73 m$^2$ and poorer cognitive performance was stronger. In the multivariable adjustment model, the MMSE and MoCA scores had ß values of -4.2 (95% CI, -5.0 to -3.4) and-5.2 (95% CI, -6.1 to -4.4), respectively. When examined as a continuous variable (per 5 ml/min/1.73 m$^2$ decrease) in crude and fully adjusted models, eGFR decrements were positively associated with lower MMSE and MoCA scores (p<0.001). In the multivariable adjustment model, the MMSE and MoCA scores had ß values of -0.4 (95% CI, -0.5 to -0.4) and -0.6 (95% CI, -0.7 to -0.5), respectively (Table 2). In all cases, lower levels of eGFR were associated with poorer cognitive performance.

To assess generalizability, a fully adjusted regression model was created for analyzing the subgroups of sex, stroke subtypes, age, SBP, NIHSS, and blood glucose levels. The results were similar in the subgroup analyses. No significant interactions were found among any of the stratified variables (male versus female, large artery versus small vessel occlusion, age, SBP, NIHSS, and blood glucose dichotomized group: higher versus lower) with eGFR or UACR for cognitive testing (Tables 3 and 4).

When combined categories of eGFR (<90 and ≥90 ml/min/1.73 m$^2$) and albuminuria (<30 and ≥30mg/g) were analyzed, the MMSE and MoCA scores were the least in those patients with UACR ≥30 mg/g and eGFR <90 ml/min/1.73 m$^2$, compared to those without albuminuria and eGFR level of ≥90 ml/min/1.73 m$^2$ in the crude and fully adjusted model. In the multivariable adjusted model, the MMSE and MoCA scores had ß values of -1.0 (95% CI, -1.5 to -0.5) and -1.7 (95% CI, -2.3 to -1.2), respectively (Table 5).

**Table 3. Multivariable adjusted* linear regression of MMSE scores with UACR and eGFR within subgroups.**

| Subgroups | Comparing UACR (30–300, >300 vs <30mg/g) | | | P for interaction | Comparing eGFR (60–90, <60 vs ≥90 ml/min/1.73m2) | | | P for interaction |
|---|---|---|---|---|---|---|---|---|
| | <30 | 30–300 | >300 | | ≥90 | 60–90 | <60 | |
| | ß (95%CI) P | | | | ß (95%CI) P | | | |
| Gender | | | | 0.050 | | | | 0.123 |
| Male | 0 | 0.3 (-0.1, 0.7) 0.137 | -0.5 (-1.5, 0.4) 0.278 | | 0 | -1.0 (-1.5, -0.5) <0.001 | -4.1 (-5.2, -3.1) <0.001 | |
| Female | 0 | -0.4 (-0.8, 0.1) 0.104 | -0.2 (-1.2, 0.8) 0.703 | | 0 | -0.5 (-1.0, -0.0) 0.050 | -4.4 (-5.7, -3.1) <0.001 | |
| Age dichotomous | | | | 0.588 | | | | 0.609 |
| Lower | 0 | -0.0 (-0.5, 0.4) 0.842 | -0.7 (-1.7, 0.4) 0.215 | | 0 | -0.8 (-1.3, -0.3) 0.002 | -5.8 (-7.4, -4.3) <0.001 | |
| Higher | 0 | 0.1 (-0.4, 0.5) 0.717 | 0.3 (-0.7, 1.3) 0.503 | | 0 | -0.8 (-1.2, -0.4) <0.001 | -3.9 (-4.6, -3.1) <0.001 | |
| Systolic blood pressure dichotomous | | | | 0.657 | | | | 0.149 |
| Lower | 0 | -0.1 (-0.5, 0.4) 0.781 | -0.2 (-1.3, 0.9) 0.740 | | 0 | -0.6 (-1.2, -0.1) 0.021 | -2.8 (-4.2, -1.4) <0.001 | |
| Higher | 0 | 0.1 (-0.3, 0.5) 0.652 | -0.3 (-1.2, 0.7) 0.562 | | 0 | -0.7 (-1.2, -0.2) 0.005 | -4.2 (-5.1, -3.3) <0.001 | |
| NIHSS score dichotomous | | | | 0.375 | | | | 0.953 |
| Lower | 0 | -0.2 (-0.7, 0.3) 0.393 | -0.6 (-1.7, 0.5) 0.304 | | 0 | -0.6 (-1.1, -0.0) 0.035 | -3.7 (-5.4, -2.0) <0.001 | |
| Higher | 0 | 0.1 (-0.3, 0.5) 0.767 | -0.0 (-0.9, 0.9) 0.982 | | 0 | -0.7 (-1.1, -0.2) 0.006 | -4.2 (-5.1, -3.3) <0.001 | |
| Stroke subtypes | | | | 0.311 | | | | 0.122 |
| Large artery | 0 | 0.1 (-0.4, 0.6) 0.661 | 0.3 (-0.7, 1.3) 0.531 | | 0 | -0.8 (-1.4, -0.2) 0.006 | -4.3 (-5.3, -3.2) <0.001 | |
| Small vessel occlusion | 0 | -0.2 (-0.6, 0.2) 0.330 | -0.4 (-1.4, 0.6) 0.441 | | 0 | -0.5 (-1.0, -0.1) 0.026 | -4.0 (-5.3, -2.8) <0.001 | |
| Blood glucose dichotomous | | | | 0.849 | | | | 0.372 |
| Lower | 0 | -0.3 (-0.9, 0.2) 0.225 | -2.5 (-5.2, 0.1) 0.059 | | 0 | -1.0 (-1.5, -0.5) <0.001 | -4.5 (-5.7, -3.4) <0.001 | |
| Higher | 0 | 0.1 (-0.4, 0.6) 0.768 | -0.0 (-1.4, 1.3) 0.951 | | 0 | -0.5 (-1.0, 0.0) 0.065 | -3.9 (-5.1, -2.8) <0.001 | |

Abbreviations: CI: confidence interval; eGFR: estimated glomerular filtration rate; UACR: urine albumin-creatinine ratio

*Adjusted, if not stratified, age, gender, BMI, education, diabetes and hypertension history, NIHSS score, stroke subtypes, smoking and alcohol drinking status, serum total cholesterol, triglyceride, uric acid, homocysteine, fasting glucose, systolic blood pressure and either estimated glomerular filtration rate or UACR.

**Table 4. Multivariable adjusted* linear regression of MoCA scores with UACR and eGFR within subgroups.**

| Subgroups | Comparing UACR (30–300, >300 vs <30mg/g) | | | P for interaction | Comparing eGFR (60–90, <60 vs ≥90 ml/min/1.73m$^2$) | | | P for interaction |
|---|---|---|---|---|---|---|---|---|
| | <30 | 30–300 | >300 | | ≥90 | 60–90 | <60 | |
| | | ß (95%CI) *P* | | | | ß (95%CI) *P* | | |
| Gender | | | | 0.107 | | | | 0.195 |
| Male | 0 | 0.3 (-0.1, 0.8) 0.149 | -0.1 (-1.1, 1.0) 0.910 | | 0 | -1.7 (-2.2, -1.1) <0.001 | -5.4 (-6.5, -4.3) <0.001 | |
| Female | 0 | -0.3 (-0.8, 0.2) 0.197 | -0.2 (-1.3, 0.8) 0.673 | | 0 | -1.1 (-1.7, -0.6) <0.001 | -4.8 (-6.2, -3.3) <0.001 | |
| Age dichotomous | | | | 0.352 | | | | 0.341 |
| Lower | 0 | 0.1 (-0.4, 0.6) 0.658 | -0.6 (-1.7, 0.6) 0.345 | | 0 | -1.3 (-1.9, -0.7) <0.001 | -6.1 (-8.0, -4.3) <0.001 | |
| Higher | 0 | 0.1 (-0.4, 0.5) 0.819 | 0.4 (-0.6, 1.4) 0.411 | | 0 | -1.4 (-1.9, -0.9) <0.001 | -5.1 (-6.1, -4.1) <0.001 | |
| Systolic blood pressure dichotomous | | | | 0.595 | | | | 0.402 |
| Lower | 0 | 0.0 (-0.5, 0.5) 0.922 | 0.1 (-1.0, 1.3) 0.825 | | 0 | -1.2 (-1.8, -0.7) <0.001 | -4.2 (-5.6, -2.7) <0.001 | |
| Higher | 0 | 0.0 (-0.4, 0.4) 0.969 | -0.3 (-1.3, 0.7) 0.533 | | 0 | -1.4 (-1.9, -0.9) <0.001 | -5.7 (-6.8, -4.6) <0.001 | |
| NIHSS score dichotomous | | | | 0.422 | | | | 0.835 |
| Lower | 0 | -0.3 (-0.8, 0.3) 0.318 | -0.1 (-1.2, 1.1) 0.932 | | 0 | -1.3 (-1.9, -0.7) <0.001 | -5.1 (-7.0, -3.3) <0.001 | |
| Higher | 0 | -0.1 (-1.2, 1.1) 0.932 | -0.1 (-1.0, 0.9) 0.879 | | 0 | -1.4 (-1.9, -0.9) <0.001 | -5.3 (-6.3, -4.4) <0.001 | |
| Stroke subtypes | | | | 0.391 | | | | 0.576 |
| Large artery | 0 | 0.3 (-0.2, 0.7) 0.277 | 0.2 (-0.8, 1.3) 0.653 | | 0 | -1.4 (-2.0, -0.8) <0.001 | -4.9 (-6.0, -3.8) <0.001 | |
| Small vessel occlusion | 0 | -0.2 (-0.7, 0.2) 0.310 | 0.1 (-1.0, 1.2) 0.825 | | 0 | -1.3 (-1.8, -0.8) <0.001 | -5.7 (-7.1, -4.2) <0.001 | |
| Blood glucose dichotomous | | | | 0.409 | | | | 0.585 |
| Lower | 0 | 0.0 (-0.4, 0.5) 0.960 | -0.5 (-1.5, 0.5) 0.303 | | 0 | -1.7 (-2.3, -1.1) <0.001 | -5.3 (-6.6, -4.0) <0.001 | |
| Higher | 0 | -0.0 (-0.5, 0.4) 0.838 | -0.1 (-1.1, 1.0) 0.918 | | 0 | -1.1 (-1.7, -0.6) <0.001 | -5.1 (-6.3, -3.9) <0.001 | |

Abbreviations: CI: confidence interval; eGFR: estimated glomerular filtration rate; UACR: urine albumin-creatinine ratio

*Adjusted, if not stratified, age, gender, BMI, education, diabetes and hypertension history, NIHSS score, stroke subtypes, smoking and alcohol drinking status, serum total cholesterol, triglyceride, uric acid, homocysteine, fasting glucose, systolic blood pressure and either estimated glomerular filtration rate or UACR.

## Discussion

This study found that in old adults with ischemic stroke in China, reduced eGFR, but not albuminuria measured by UACR, has an independent association with poorer cognitive performance. This study supports findings from prior studies reporting a link between kidney function and cognitive impairment.

### eGFR and cognitive performance

Our results showed that, in the absence or presence of albuminuria, an eGFR <690 ml/min/1.73 m$^2$ had an association with worse cognitive performance. However, the presence of

**Table 5. Association of joint UACR and eGFR stratum with cognitive performance (β 95% CI).**

| eGFR, (ml/min/1.73m$^2$) | | MMSE | | MoCA | |
|---|---|---|---|---|---|
| | | UACR(mg/g) categories | | UACR(mg/g) categories | |
| | | <30 | ≥30 | <30 | ≥30 |
| ≥90 | Crude | 0.00 | 0.0 (-0.4, 0.4) | 0.00 | 0.2 (-0.3, 0.6) |
| | Adjusted* | 0.00 | -0.0 (-0.4, 0.4) | 0.00 | 0.1 (-0.3, 0.6) |
| <90 | Crude | -1.2 (-1.7, -0.8) | -1.5 (-2.0, -1.0) | -1.7 (-2.2, -1.3) | -2.1 (-2.6, -1.6) |
| | Adjusted* | -0.8 (-1.3, -0.4) | -1.0 (-1.5, -0.5) | -1.4 (-1.9, -0.9) | -1.7 (-2.3, -1.2) |

Abbreviations: CI: confidence interval; eGFR: estimated glomerular filtration rate; UACR, urine albumin-creatinine ratio.

* adjusted for age, gender, BMI, education, diabetes and hypertension history, NIHSS score, stroke subtypes, smoking and alcohol drinking status, serum total cholesterol, triglyceride, uric acid, homocysteine, fasting glucose, systolic blood pressure.

albuminuria amplified the relationship. These findings corroborate the findings of prior studies on the association between renal measures and poorer cognitive performance. According to previous reports, this association exists during the early stages of renal dysfunction [17, 18]. The association between renal function and cognitive performance has been paid much attention in recent decades. Wang et al. found that both CKD and eGFR were associated with cognitive dysfunction among older patients with hypertension in China [32]. A prospective study demonstrated that kidney dysfunction is associated with a cognitive decline in a community-based population, and this association is independent of albuminuria [33]. Another longitudinal study suggested that a low eGFR was associated with increased risk of dementia. In addition, a faster decrease in eGFR and proteinuria was associated with incidence of cognitive impairment and dementia; this association may be mediated by vascular mechanisms [20]. Additionally, a large population study found that each 10 ml/min/1.73 m$^2$ decrease in eGFR<60 ml/min/1.73 m$^2$ was independently associated with an increased prevalence of cognitive impairment [34]. The OSACA2 study found that CKD was independently related to incident dementia in patients with vascular risk factors [27]. However, one cohort study from the Oxford Vascular Study (OXVASC) found that CKD was not associated with either pre- or post-event dementia in patients with TIA and stroke [28]. The inconsistent results may be attributable to differences in the study populations and the severity of lesions. OXVASC focuses on the relationship between CKD and dementia in patients with TIA and stroke, while the participants in SPRINT were older community-dwelling adults with cardiovascular disease. Moreover, OSACA2 included high-risk patients for the primary and secondary prevention of cardiovascular disease. Another potential reason could be the different methods of assessment. Many previous studies have used different assessment tools to evaluate cognitive function.

Although the precise pathophysiologic mechanisms that underlie the association between low eGFR and poorer cognitive performance are not known and need further studies for clarity, several factors may help explain the association between low eGFR and cognitive dysfunction. Patients with renal dysfunction have a higher prevalence of cerebrovascular disease (CVD) than the general population. Renal dysfunction may contribute to vascular cognitive impairment through traditional CVD risk factors, including hypertension, diabetes mellitus, dyslipidemia, smoking, and other non-traditional vascular risk factors comprising hyperhomocysteinemia, hyperuricemia, anemia, inflammation, and oxidative stress. Moreover, CKD characterized by the accumulation of uremic toxins, excessive RAS activation, renal anemia, and renal osteopathy, caused by renal dysfunction are possible mechanisms of renal disease involvement in the progression of dementia [35, 36]. Our published report demonstrated that low eGFR may mediate the association between SUA and cognitive impairment in patients with ischemic stroke [37]. Conceptually, low eGFR may lead to the accumulation of neurotoxins or may represent a prolonged exposure to CVD or its risk factors [6]. Vascular dementia is often linked to hypertension, which is a common finding in CKD. Hypertension has often been associated with small lacunar infarcts and leukoaraiosis, cerebral autoregulation dysfunction, and chronic hypoperfusion. Although cerebral blood flow may be adequate in CKD, the blood–brain barrier or the glymphatic system might be dysfunctional. In this study, more than 50% of the patients had hypertension. Uremic toxins in circulation, which enter the central nervous system (CNS) as a result of blood-brain barrier damage during kidney dysfunction, might impair CNS structure and function [38].

## Albuminuria and cognitive performance

According to recent systematic review and meta-analysis studies, albuminuria has an independent association with increased risk of cognitive impairment and dementia [4, 39, 40]. To the

best of our knowledge, albuminuria is a significant risk factor for the development of dementia in community-based Japanese older adults. Moreover, low eGFR and albuminuria are mutually associated with a greater risk of vascular dementia [41]. A prospective cohort study demonstrated that persistent or progressive albuminuria is associated with cognitive decline in adults with diabetes and normal eGFR [42]. Wei et al. found that urine albumin is significantly associated with cognitive dysfunction, and can be potentially used as an early marker for CKD [43].

The mechanism underlying the association between albuminuria and poorer cognition is not completely understood and is likely multifactorial. Both cerebral and renal circulation are characterized by high flow and low impedance. The association between albuminuria and cognitive impairment probably reflects common microvascular pathogenesis in the kidneys and brain. Several reasons may explain the inconsistencies between our findings and those of previous reports. First, the conflicting results may be attributable to use of different study populations in studying the severity of albuminuria. Cognitive dysfunction in patients may be indicative of another chronic microvascular disease. This may explain why dementia and poorer cognitive performance are more common in individuals with diabetes who have a high prevalence of albuminuria [42, 44]. However, since fewer patients with diabetes participated in this study (8%), these findings should be interpreted with caution. The study might be underpowered to detect the association between UACR and poorer cognitive performance. Second, this study only measured UACR at a single time point. UACR is affected by exercise and the time of day the sample is taken [45], which is why repeated measurements would be preferred. The true association between albuminuria and cognitive performance is likely to be diluted. An alternative explanation for the disparate results of the association between low eGFR and albuminuria and poorer cognitive performance could be that albuminuria is a marker for a different condition other than kidney disease [46].

This study had several strengths. First, the primary independent and dependent variables were quantified using clinically relevant measures (eGFR and UACR for two independent markers of renal function and damage and MoCA and MMSE for cognitive performance), thus providing clinical relevance to the analyses. In addition, the use of both eGFR and UACR as correlators enabled us to test their independent and joint effects on cognitive performance. Second, the study controlled for several potential confounders, including demographic and clinical parameters, while assessing the association.

Several limitations were noted. First, our study population comprised a small number of patients with CKD (as defined by eGFR $<60$ ml/min/1.73 m$^2$), and this limited our ability to detect a relationship between eGFR $<60$ ml/min/1.73 m$^2$ and cognitive performance. Second, these parameters were based on a one-time evaluation and do not provide information on the duration or severity of risk factors over time. Third, we did not analyze impairment in specific cognitive domains; instead, overall cognitive scores were used in the analysis. Fourth, the cognitive assessment was performed one month after hospital discharge, and the incidence of post-stroke cognitive impairment has been reported to be the highest at three months after stroke [47]. Therefore, there is a need for further long-term follow-up. Fifth, although older age is the most significant risk factor for cognitive decline, patients above 80 were not included in our study. This is due to the small number of oldest-old patients and the difficulty with outpatient follow-up. Finally, we acknowledge the possibility of residual confounding by other unmeasured covariates, such as cystatin C.

Our findings support the hypothesis that renal dysfunction and albuminuria may operate through different pathophysiologic mechanisms. They further suggest that the pathways mediating the association between low eGFR and albuminuria and poorer cognitive performance may be partially distinct. The findings have clinical implications. Our results suggest that the

measurement of eGFR is a good biomarker for cognitive impairment in patients with acute ischemic stroke. Low eGFR may reflect a partially distinct pathophysiologic process of cognitive dysfunction. Targeting the pathogenesis of low eGFR or treating modifiable risk factors may have clinical implications in preventing cognitive impairment or delaying disease progression after stroke. This would imply that mild reductions in kidney function may be accompanied by cerebral vascular changes that predispose patients to altered cognitive performance. Further studies are needed to confirm whether the setting of kidney dysfunction is a possible new determinant for cognitive impairment in patients with stroke.

## Conclusion

Our findings suggest that in patients with ischemic stroke, reduced eGFR but not albuminuria was associated with cognitive impairment, as measured by MMSE or MoCA. The eGFR decline could be an effective indicator of cognitive impairment after a stroke. These results suggest that the early stages of renal dysfunction could be an effective target to prevent cognitive impairment in patients with stroke. It emphasizes the need and possibility for early detection and early prevention of cognitive impairment after stroke.

## Supporting information

**S1 Data.**
(XLSX)

## Author Contributions

**Conceptualization:** Chunyan Zhang, Dongfang Li.

**Data curation:** Yanjuan Hou, Huizhong Gao, Bo Bai.

**Formal analysis:** Chunyan Zhang.

**Funding acquisition:** Chunyan Zhang, Pengfei Meng.

**Investigation:** Yanjuan Hou, Huizhong Gao, Bo Bai.

**Methodology:** Pengfei Meng.

**Project administration:** Chunyan Zhang, Dongfang Li.

**Resources:** Huizhong Gao.

**Software:** Pengfei Meng.

**Supervision:** Guofang Xue.

**Validation:** Pengfei Meng.

**Visualization:** Guofang Xue, Yanjuan Hou.

**Writing – original draft:** Chunyan Zhang, Bo Bai.

**Writing – review & editing:** Chunyan Zhang, Guofang Xue, Dongfang Li.

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
