## [Decision Letter · Decision Letter 0]

24 Oct 2023

PONE-D-23-30196Association between kidney measurements and cognitive performance in patients with ischemic strokePLOS ONE

Dear Dr. Zhang,

Thank you for submitting your manuscript to PLOS ONE. After careful consideration, we feel that it has merit but does not fully meet PLOS ONE’s publication criteria as it currently stands. Therefore, we invite you to submit a revised version of the manuscript that addresses the points raised during the review process.

Please address the comments by the 2 reviewers and resubmit

We look forward to receiving your revised manuscript.

Kind regards,

Yee Gary Ang, MBBS MPH

Academic Editor

PLOS ONE

Journal Requirements:

Reviewers' comments:

Reviewer's Responses to Questions

**Comments to the Author**

1. Is the manuscript technically sound, and do the data support the conclusions?

Reviewer #1: Yes

Reviewer #2: Yes

2. Has the statistical analysis been performed appropriately and rigorously? 

Reviewer #1: Yes

Reviewer #2: Yes

3. Have the authors made all data underlying the findings in their manuscript fully available?

Reviewer #1: Yes

Reviewer #2: Yes

4. Is the manuscript presented in an intelligible fashion and written in standard English?

Reviewer #1: Yes

Reviewer #2: Yes

5. Review Comments to the Author

Reviewer #1: This is a scientfically sound manuscript. Thanks for the opportunity to review.

I have just a few comments.

1. As far as I can see patients above 80 are not included. Since the geriatric population is becoming more and more important, could you please add a rationale for this.

2. The univariate and the multivariate analyses are presented with a similar importance. Could you please put a bit more emphasis on the multivariate analysis, since this is more reliable.

3. Please add unmeasured confounding as possible limitation.

Reviewer #2: Thank you for this interesting and well written paper. I have a few comments I hope will assist with strengthening your paper further.

In the abstract your first sentence states the relationships here are less clear - it is not apparent what they are less clear than. Please re-phrase or include a comparison to show where the other relationship you refer to has been better clarified.

In your results in some areas you refer to associations existing between your variables but do not describe the directionality of this association. It would be conventional to be explicit about whether this is a positive or negative association and briefly state what this means for the reader. Page 12 lines 216-217 is one example.

My major comment is just around the importance of this paper - the impact of these poorer renal markers on cognitive performance is well understood, as is the increased potential of worse renal function in those who suffer from ischemic stroke, so it is typically understood that those with ischemic stroke are more at risk of cognitive decline. I think you need a stronger message to support your paper and show why your work is novel in this context, or further push the message of the importance of this knowledge and early intervention in a clinical setting.

6. PLOS authors have the option to publish the peer review history of their article (what does this mean?). If published, this will include your full peer review and any attached files.

Reviewer #1: No

Reviewer #2: No

---

## [Author Response · Author response to Decision Letter 0]

15 Nov 2023

RE: Manuscript Number PONE-D-23-30196

Title: Association between kidney measurements and cognitive performance in patients with ischemic stroke

Dear editor,

We are pleased to resubmit a revised version of our manuscript entitled “Association between kidney measurements and cognitive performance in patients with ischemic stroke” to the Journal of PLOS ONE. We greatly appreciate the careful review and comments from you and the reviewers and we believe that we now have a stronger manuscript. We have responded to each of these comments and have revised our manuscript accordingly.

Updated statement: This study was supported by the Scientific Research Foundation for Doctors, the Second Hospital of Shanxi Medical University, China (Grant No: 201601-9), Natural Science Foundation of Shanxi Province (Grant No: 201801D221411 and 20210302124427), Science and Technology Department of Shanxi Province, China. Funders played a role in the collection, analysis, and preparation of the research data. None of the authors have received a salary from the funders.

We thank you again for giving us the opportunity to revise this manuscript.

Yours sincerely,

Chunyan Zhang

Department of Neurology, Second Hospital, Shanxi Medical University, 382 Wuyi Road, Taiyuan, Shanxi, China 

Phone: 86-351-3365204

Email: marzipanzcy@126.com

Detailed Responses to the Reviewers’ Comments

Reviewer #1: 

Comments: 

Issue 1

Question: As far as I can see patients above 80 are not included. Since the geriatric population is becoming more and more important, could you please add a rationale for this.

Response:

We have addressed this limitation in the revised manuscript (Page 19, lines 358–360) as follows: 

Although older age is the most significant risk factor for cognitive decline, patients above 80 were not included in our study. This is due to the small number of oldest-old patients and the difficulty with outpatient follow-up.

Issue 2

Question: The univariate and the multivariate analyses are presented with a similar importance. Could you please put a bit more emphasis on the multivariate analysis, since this is more reliable.

Response: According to the reviewer’s suggestion, we have updated the results section and emphasized the results of multivariate analyses (Page 13 and 14, lines 234–250).

Issue 3

Question：Please add unmeasured confounding as possible limitation.

Response: As suggested by the reviewer, we have addressed this limitation in the revised manuscript (Page 19-20, lines 360–362) as follows:

Finally, we acknowledge the possibility of residual confounding by other unmeasured covariates, such as cystatin C.

Reviewer #2: 

Comments: 

Issue 1

Question：In the abstract your first sentence states the relationships here are less clear - it is not apparent what they are less clear than. Please re-phrase or include a comparison to show where the other relationship you refer to has been better clarified.

Response: As suggested, we have re-phrased the background section of the abstract as follows (Page 2, lines 21–28).

Individuals with chronic kidney disease (CKD) are at a substantially higher risk for stroke, which may predispose individuals to cognitive impairment. However, the association of low estimated glomerular filtration rate (eGFR) and albuminuria with poorer cognitive performance in patients with stroke is not fully understood, and the current evidence for this association is contradictory. Our aim was to retrospectively investigate whether low eGFR and albuminuria, as indicated by the urine albumin-creatinine ratio (UACR), are independently or jointly associated with worse cognitive performance in patients with ischemic stroke.

Issue 2

Question：In your results in some areas you refer to associations existing between your variables but do not describe the directionality of this association. It would be conventional to be explicit about whether this is a positive or negative association and briefly state what this means for the reader. Page 12 lines 216-217 is one example.

Response: We have updated the results section to clarify the directionality of the associations between renal measurements and cognitive performance (Page 12 and 13, lines 229–232).

Issue 3

Question：My major comment is just around the importance of this paper - the impact of these poorer renal markers on cognitive performance is well understood, as is the increased potential of worse renal function in those who suffer from ischemic stroke, so it is typically understood that those with ischemic stroke are more at risk of cognitive decline. I think you need a stronger message to support your paper and show why your work is novel in this context, or further push the message of the importance of this knowledge and early intervention in a clinical setting.

Response: The reviewer raises a good point. In order to emphasize the importance of our work, we have updated the background section to explain why we studied this association and have added new supporting citations (Page 5, line 85 to Page 6, line 112).

Post-stroke cognitive impairment (PSCI) is common after stroke and occurs in up to 60% of stroke survivors in the first year after stroke, with a higher rate seen shortly after stroke. (1) The clinical determinants of PSCI are not fully understood. Various putative risk factors have been inconsistently reported, including older age, risk factors for cerebrovascular disease (e.g., hypertension, diabetes mellitus, and atrial fibrillation), stroke features, and lesion characteristics. (2) The kidney and the brain have strong similarities in vascular organization, and a complex interplay has been found between CKD and cerebrovascular disease.(3) Individuals with CKD are at a substantially higher risk for stroke, which may predispose individuals to cognitive impairment. However, only a few studies have examined the association of low eGFR and albuminuria with poorer cognitive performance in individuals at high risk or patients with stroke, and the current evidence for such an association is contradictory.

The identification of modifiable risk factors of cognitive decline after stroke is of vital importance in early preventive and intervention strategies for PSCI.(1, 4) Furthermore, kidney deficits may have a potential association with stroke and post-stroke cognitive impairment. Therefore, this study aimed to investigate whether low eGFR and albuminuria, as measured by UACR, are independently or jointly associated with worse cognitive performance in patients with ischemic stroke.

Additional references:

1. El Husseini N, Katzan IL, Rost NS, Blake ML, Byun E, Pendlebury ST, et al. Cognitive Impairment After Ischemic and Hemorrhagic Stroke: A Scientific Statement From the American Heart Association/American Stroke Association. Stroke. 2023 Jun;54(6):e272-e91. PubMed PMID: 37125534. Epub 2023/05/01. eng.

2. Lo JW, Crawford JD, Desmond DW. Profile of and risk factors for poststroke cognitive impairment in diverse ethnoregional groups. 2019 Dec 10;93(24):e2257-e71. PubMed PMID: 31712368.

3. Hanna RM, Ferrey A, Rhee CM, Kalantar-Zadeh K. Renal-Cerebral Pathophysiology: The Interplay Between Chronic Kidney Disease and Cerebrovascular Disease. Journal of stroke and cerebrovascular diseases : the official journal of National Stroke Association. 2021 Sep;30(9):105461. PubMed PMID: 33199089. Epub 2020/11/18. eng.

4. Drew DA, Weiner DE, Sarnak MJ. Cognitive Impairment in CKD: Pathophysiology, Management, and Prevention. American journal of kidney diseases : the official journal of the National Kidney Foundation. 2019 Dec;74(6):782-90. PubMed PMID: 31378643. Pubmed Central PMCID: Pmc7038648. Epub 2019/08/06. eng.

---

## [Decision Letter · Decision Letter 1]

4 Dec 2023

Association between kidney measurements and cognitive performance in patients with ischemic stroke

PONE-D-23-30196R1

Dear Dr. Zhang,

We’re pleased to inform you that your manuscript has been judged scientifically suitable for publication and will be formally accepted for publication once it meets all outstanding technical requirements.

Kind regards,

Yee Gary Ang, MBBS MPH

Academic Editor

PLOS ONE

Additional Editor Comments (optional):

Reviewers' comments:

Reviewer's Responses to Questions

**Comments to the Author**

1. If the authors have adequately addressed your comments raised in a previous round of review and you feel that this manuscript is now acceptable for publication, you may indicate that here to bypass the “Comments to the Author” section, enter your conflict of interest statement in the “Confidential to Editor” section, and submit your "Accept" recommendation.

Reviewer #1: All comments have been addressed

Reviewer #2: All comments have been addressed

2. Is the manuscript technically sound, and do the data support the conclusions?

Reviewer #1: Yes

Reviewer #2: Yes

3. Has the statistical analysis been performed appropriately and rigorously? 

Reviewer #1: Yes

Reviewer #2: Yes

4. Have the authors made all data underlying the findings in their manuscript fully available?

Reviewer #1: Yes

Reviewer #2: Yes

5. Is the manuscript presented in an intelligible fashion and written in standard English?

Reviewer #1: Yes

Reviewer #2: Yes

6. Review Comments to the Author

Reviewer #1: (No Response)

Reviewer #2: Thank you for addressing the comments made. This is a well written paper and interesting analysis. I have no further comments

7. PLOS authors have the option to publish the peer review history of their article (what does this mean?). If published, this will include your full peer review and any attached files.

Reviewer #1: **Yes: **Martin Gebel

Reviewer #2: No

---

## [Editor Report · Acceptance letter]

5 Dec 2023

PONE-D-23-30196R1 

Association between kidney measurements and cognitive performance in patients with ischemic stroke 

Dear Dr. Zhang:

I'm pleased to inform you that your manuscript has been deemed suitable for publication in PLOS ONE. Congratulations! Your manuscript is now with our production department. 

Kind regards, 

on behalf of

Dr. Yee Gary Ang 

Academic Editor

PLOS ONE